# Rheumatoid Factor: A Novel Determiner in Cancer History

**DOI:** 10.3390/cancers13040591

**Published:** 2021-02-03

**Authors:** Alessio Ugolini, Marianna Nuti

**Affiliations:** 1Department of Experimental Medicine, “Sapienza” University of Rome, Viale Regina Elena 324, 00161 Rome, Italy; alessio.ugolini@moffitt.org; 2Department of Immunology, H. Lee Moffitt Cancer Center, Tampa, FL 33612, USA

**Keywords:** rheumatoid factor, autoimmunity, autoantibodies, cancer, biomarker, predictive biomarker, prognostic biomarker, cancer progression, cancer development, immunotherapy, cancer susceptibility, tumor recurrence, tumor load

## Abstract

**Simple Summary:**

Rheumatoid factors are autoantibodies that characterize different autoimmune diseases, in particular rheumatoid arthritis, but that can also be found in the sera of the general healthy population. They have been mainly studied in the context of autoimmune diseases, but some evidence have suggested an association between their presence and the predisposition to develop cancer as well as a facilitation of cancer growth and progression in oncologic patients. In this review, for the first time we thus analyze and discuss the possible roles that these autoantibodies can assume in tumor history, from determiners of a heightened susceptibility of developing cancer to drivers of a reduced response to immunotherapies.

**Abstract:**

The possible interplay between autoimmunity and cancer is a topic that still needs to be deeply explored. Rheumatoid factors are autoantibodies that are able to bind the constant regions (Fc) of immunoglobulins class G (IgGs). In physiological conditions, their production is a transient event aimed at contributing to the elimination of pathogens as well as limiting a redundant immune response by facilitating the clearance of antibodies and immune complexes. Their production can become persistent in case of different chronic infections or diseases, being for instance a fundamental marker for the diagnosis and prognosis of rheumatoid arthritis. Their presence is also associated with aging. Some studies highlighted how elevated levels of rheumatoid factors (RFs) in the blood of patients are correlated with an increased cancer risk, tumor recurrence, and load and with a reduced response to anti-tumor immunotherapies. In line with their physiological roles, RFs showed in different works the ability to impair in vitro anti-cancer immune responses and effector functions, suggesting their potential immunosuppressive activity in the context of tumor immunity. Thus, the aim of this review is to investigate the emerging role of RFs as determiners of cancer faith.

## 1. Introduction

The link between autoimmunity and cancer is considered a hot topic since the relationship existing between these two conditions is still to be clarified. Rheumatoid factors (RFs) are autoantibodies with different isotypes and affinities, which bind the constant regions (Fc) of immunoglobulins class G (IgG). RFs were initially discovered in sera of patients with rheumatoid arthritis (RA), and they are still considered a fundamental marker for the diagnosis and prediction of the prognosis of these patients [1,2]. Later, it was highlighted that RFs are not crucial for the development of the arthritis and, above all, that they are not specific only for RA [3].

In fact, high levels of RFs can be found in the sera of patients with other diseases (both autoimmune and non-autoimmune) in the same way as in healthy subjects [4].

## 2. Rheumatoid Factors Isotypes and Affinities

RFs mainly belong to the immunoglobulins of class M (IgM), class G (IgG), and class A (IgA) isotypes; rarely, also class E (IgE) and class D (IgD) RFs can be detected. While physiological RFs are mainly of the IgM isotype, are polyreactive, have low affinity, and show a reduced usage of the V gene (encoding for the variable region of the antibody), pathological RFs can belong to IgM, IgA, IgG, IgD, and IgE classes and show a high affinity and a wide usage of the V gene, thus indicating that pathological RFs are the result of an immune response against a specific antigen [5,6,7,8,9].

RFs can bind to the Fc of all four subclasses of IgG. In different studies mapping the RFs’ binding sites, it resulted that the affinity for the IgG1 subclass was the highest, whereas the affinity for the IgG3 subclass was the most variable among the different sera samples that were tested [10,11].

## 3. Physiological Rheumatoid Factor Production and Its Presence in Different Clinical Conditions

In peripheral blood of healthy subjects, researchers found a B-cells repertoire that was able to secrete RFs (RF^+^ B-cells) [12,13,14]. This population seems to be anergic in subjects that are RF-seronegative, while it requires a specific activation pattern to start synthetizing RFs [15].

Whereas in pathological conditions, such as RA, the chronic presence of RFs in patient sera is due to the production carried out by terminally differentiated plasma cells in the absence of a specific stimulus [16,17], in physiological conditions, the RFs production is a transient event that results from an initiating stimulus capable of activating the B-cells repertoire [15]. This initiating stimulus can be represented by an infection (bacterial, viral, or parasitic) or by an active immunization [18,19,20,21,22,23]. The activation of this repertoire of RF^+^ B-cells is due to their interaction with T-helper cells, which react against a foreign antigen during a secondary immune response [24,25]. In fact, it was proven that activated T-cells are strong inducers of RF^+^ B-cells and, therefore, of physiological RFs production [12,13,14,15]. This is coherent with the RFs physiological role: on one side, they seem essential in fighting pathogens by contributing to the formation and clearance of immune complexes (thanks to IgM and IgG RFs capability at forming bigger immune complexes that can both bind the complement and be phagocytosed) and because RF^+^ B-cells can act as antigen-presenting cells (APCs); on the other side, RFs are also important in limiting a redundant immune response against pathogens by destroying the antibodies produced in excess [26,27,28,29].

### 3.1. RFs in Patients with Non-Autoimmune Conditions

As outlined above, RFs production is essential in protecting the host against infections in an inflammatory milieu. This is why high levels of RFs can be detected in patients with different types of infections and chronic diseases. Conversely to the RFs found in RA, those detected during infections are not damaging and are usually transient [4]. They are also, as physiological RFs, polyreactive and low-affinity IgMs that show a reduced usage of V gene [6,7,9]. If the infection evolves in a chronic disease, also the RFs circulating levels can become persistent. RFs can indeed be found in the sera of 40–50% of patients with HCV infection, reaching even 76% in some studies [30]. This is probably due to a continuous stimulation and activation of an immune response triggered by the presence of the virus in HCV patients. Since HCV infection nowadays has reached a high prevalence in a large number of countries, it has become the first cause of high RFs levels in sera [30,31].

### 3.2. RFs in the General Healthy Population

High levels of RFs can be detected in the general healthy population, with a worldwide variability in prevalence: for example, RFs positivity in sera show the highest prevalence in North American Indian tribes (up to 30%), while in young Caucasians it is up to 4% [32,33,34,35,36,37]. As physiological RFs, those detected in the healthy population are not damaging, are usually transient, and are polyreactive and low-affinity IgM, showing a reduced usage of the V gene [4,6,7,9]. The transient production of these physiological RFs can be the result of any kind of infection [18,19,20,21,22,23,38]. Polyreactive IgM RFs can be persistently found in 18% of presumably healthy aging subjects, suggesting that their chronic production could be an age-related immune deregulation phenomenon [,[39],[40],[41]], whereas in other individuals, the reason of their presence can still not be identified [4].

### 3.3. RFs in Rheumatoid Arthritis and Other Autoimmune Diseases

In the sera of RA patients, IgM is the most frequent RFs isotype detected, which is followed by IgG and IgA and, very rarely, also IgE and IgD. Most (70–90%) RA patients are RF-positive; three isotypes of RFs, IgM, IgG, and IgA, are detected in up to 52% of patients with RA [4,5,8,42,43,44,45]. RFs can also be found in the sera of patients with other autoimmune systemic syndromes, such as Sjogren’s syndrome (SS), mixed cryoglobulinemia, systemic lupus erythematosus (SLE), mixed connective tissue disease, polymyositis, and dermatomyositis. They can be detected also in 10% of patients with Waldenstrom’s macroglobulinemia (a rare plasma cell cancer). Patients with SS (up to 60%) and type II and type III of mixed cryoglobulinemia (often HCV-related) show the highest RFs titers [26,43,46,47,48,49].

## 4. Rheumatoid Factor and Cancer History

During the years, the presence of circulating RFs was almost exclusively correlated with the diagnosis and prognosis of RA and other autoimmune diseases. The role of RFs in cancer was poorly investigated.

The presence of RFs can be detected in the blood of 10–20% of cancer patients [50], reaching 26% in non-small lung cancer (NSCLC) patients [51]. The higher prevalence of RF positivity in cancer patients compared to the general healthy population can be surely explained by the older age of subjects affected by cancer, since RFs production is associated with aging; however, it could also suggest a possible association between RF positivity and cancer. In this scenario, its production could be the result of a regulatory-skewing B-cells activation.

This association was investigated for the first time in the Reykjavik area (Iceland) where, starting from 1967, a general health survey was conducted [52,53]. Then, the women that were tested positive for RF were divided in groups based on RF titers and followed up until 1974: of the four women who died during this observation period, three belonged to the group with the highest RF titers; all of these three women had been diagnosed with cancer (two mammalian cancer and one lung cancer). Thus, it was suggested that high blood concentrations of RFs in healthy subjects might be associated with an increased risk of developing cancer [52,53].

After this study, other publications showed how the presence of high RF titers in patient sera were associated with an increased cancer risk, tumor recurrence, and tumor load if compared with patients that were RF-negative [54,55,56,57,58,59,60].

In particular, in a longitudinal study conducted in 2016 including 2331 patients with early RA, Ajeganova et al. [54] studied the presence of RFs, anticitrullinated protein antibodies (ACPA), and anticarbamylated protein (anti-CarP) antibodies in relation to all causes of mortality. Interestingly, they found that the presence of RF, differently from the other autoantibodies, was associated with an increased number of neoplasm-related deaths [54].

In a cohort study made in 2017 and involving 295,837 RA-free participants, Ahn et al. clearly showed how cancer mortality risk was significantly greater in healthy adults that were positive for RF when compared with those that were RF-negative; moreover, they also demonstrated that cancer mortality risk was even higher in subjects with RF titers greater than 100 IU/mL than in those with RF-negativity, suggesting a dose-dependent effect [55].

Finally, a retrospective study conducted in our laboratory brought clear evidence of how the IgM–RF positivity is a strong predictive factor for the development of NSCLC patients’ early progression in response to the treatment with anti-PD-1 immune checkpoint inhibitors (ICIs) [51,61]. IgM-RF also correlates with a negative prognosis in terms of both overall survival (OS) and progression-free survival (PFS) in metastatic NSCLC patients in treatment with an anti-PD-1 ICI, with the worst outcome shown by patients with titers greater than 50 IU/mL [51,61].

Taken together, all these studies strongly suggest a facilitation of cancer growth and progression in patients that are positive for RFs. The mechanism lying behind this phenomenon still remains unknown, but some in vitro experiments highlighted an association between the presence of RFs and an altered anti-tumor immunity.

Indeed, it was pointed out that RF preparations are able to impair the tumor-specific in vitro cytotoxicity of cancer patients’ lymphoid cells [62,63]; IgM preparations lacking of RF anti-IgG activity, used as control, did not block the cytotoxicity, indicating that the impairing effect was the result of the specific RFs activity. The pre-incubation of lymphoid effector cells with the RF preparations inhibited their cytotoxic action, whereas pre-incubation of the tumor target cells with RF preparations before cytotoxic lymphocytes were added had no effect, thus indicating that the observed phenomenon was mediated by a direct effect on lymphoid effector cells [62].

Another important evidence supporting the suppressive activity of RFs on lymphoid effector cells directed against tumor cells came out when it was shown that RF can have a blocking effect in antibody-dependent cell-mediated cytotoxicity (ADCC) [64,65], which is an important mechanism underlying the killing of tumor cells exerted by lymphoid cells.

In addition, Giuliano et al. [66] demonstrated that RFs are able to affect the melanoma patients’ humoral immune response directed against membrane antigens of melanoma cells in vitro. They indeed showed that the presence of RF in Indirect Membrane Immunofluorescence (IMI) assays increases the IgM reactivity detection, while in the Immune Adherence (IA) assays, its presence reduces the detection of anti-membrane antibodies, thus suggesting that the presence of RF prevents the binding of anti-tumor antibodies to their target antigens on cancer cells [66].

Interestingly, in 2013, Jones at al. [67] found that RF can inhibit Rituximab effector function. Rituximab is an IgG1 monoclonal antibody directed against the receptor CD20, expressed by B-cells, which uses the complement-dependent cytotoxicity (CDC) and other mechanisms to eliminate pathogenic B-cells [68]. It is used in the treatment of some B-cell neoplasms, RA, and other autoimmune diseases [69,70]. In this study, Jones et al. demonstrated that RF inhibits Rituximab-mediated CDC. Since RF does not block the interaction between Rituximab and B-cells, it seems plausible that RF impairs this effector function through the recognition and the binding of Rituximab Fc, which mediates the CDC. Supporting this, they demonstrated that RF can also inhibit the trogocytosis, which is an FcγR-dependent effect [67].

In accordance with these observations, in a recent work, we showed that IgM-RF is not able to prevent the engagement between the drug Nivolumab (an IgG4 monoclonal antibody) and its target receptor PD-1 on T-cells [51]. Instead, IgM-RF is able to bind preferentially naïve and central memory CD4^+^ and CD8^+^ T-cells, leading to an impaired in vitro migration of these T-cell subsets in response to the CCL19 cytokine [51,61]. Moreover, RF-positive NSCLC patients showed a significant reduction of the CD137^+^ T-cells, which identify the tumor-specific effector T-cell population [71]. This suggests that the dysfunctional recirculation of naïve and central memory T-cells due to the presence of IgM-RF can lead to an impaired expansion of the tumor-directed effector T-cell population, consequently resulting in the failure of the anti-PD-1 treatments that relies on tumor-specific effector T-cells in order to be effective [51] (Figure 1).

## 5. Conclusions

Altogether, these findings suggest that high titers of RF in patients’ sera could inhibit the anti-tumor immunity in different ways (Figure 2).

In fact, RFs physiological role of limiting a redundant immune response against pathogens seems coherent with their potential immunosuppressive activity within anti-tumor immune responses. On one side, in healthy conditions, it was pointed out how RFs are able to facilitate the clearance of immune complexes and antibodies. Similarly, in cancer, their ability to bind antibodies results in interfering with the anti-tumor effect of both endogenous and therapeutic antibodies. Indeed, it has been demonstrated that RFs can hamper the interaction between tumor-directed antibodies and antigens on cancer cells’ surface as well as impair antibodies-mediated anti-tumor effector functions, such as CDC and ADCC. On the other side, the physiological effect exerted by RFs on effector T-cells has not been clarified yet. However, from different works presented in this review, it emerged that in the context of anti-cancer immune responses, the presence of RFs is able to impair lymphocytes-mediated anti-tumor cytotoxicity and the recirculation of naïve and central memory T-cells, leading to a reduced expansion and effectiveness of tumor-specific effector T-cells. Therefore, in the light of these results, it is reliable to assume that RFs’ presence in the blood of cancer patients could facilitate cancer growth and progression.

However, although some evidence of the pro-tumor role of RFs was provided in vitro, in vivo data are still lacking.

Thus, the altered anti-tumor immunity due to the presence of RFs may lead both to an increased risk of developing cancer in RF-positive subjects and to a failure of immune-based anti-tumor therapies, in terms of a higher amount of early progressions and a reduced overall survival and progression-free survival rate following immunotherapies. Different studies have indeed demonstrated the association between RF positivity and the predisposition to develop cancer, the increased tumor recurrence and tumor load, the worse prognosis, and eventually the failure of immune checkpoint inhibitors therapies in cancer patients.

These findings opened the possibility of using RFs as both prognostic and predictive biomarkers in cancer patients, which looks promising from two different perspectives. First, being now clear that RFs can be used as an indicator of a heightened susceptibility of developing cancer as well as of an increased tumor recurrence, subjects that are positive for RFs could be monitored more strictly so as to increase early cancer diagnosis. Second, since the presence of RFs in cancer patients correlates with a reduced OS and PFS and a lack of response to anti-cancer immunotherapies, the RFs’ dosage could be introduced in oncologic patients together with other parameters in order to improve their risk stratification and help adjust the therapeutic plan based on the single patient’s characteristics. Since RF dosage is a routinely performed and already standardized test, this seems to be a easily practicable and promising opportunity to aid clinicians that are struggling to predict patients faith and, above all, to increase the number of responder patients to anti-cancer therapies.

Finally, the higher prevalence of RFs in cancer patients compared to healthy adults could certainly be referred to the older age of subjects affected by cancer, but it could also be more evidence of the association between RF positivity and cancer risk. Indeed, even if direct proof is lacking, we would like to take into consideration the hypothesis that RF secretion could be the result of a regulatory-skewing B-cells activation due to the chronic inflammation carried out by the presence of the tumor itself and, in this scenario, it could serve as a further mechanism exerted by the tumor in its inflammatory milieu in order to escape from the immune surveillance.

Other studies will be required to further clarify the role of RFs in cancer. Nevertheless, the aim of this review is to open a new chapter in the study of cancer history, a chapter in which RF can be considered as a novel determiner of cancer susceptibility and a novel predictive and prognostic biomarker of a negative outcome in cancer patients that should now be taken into account when stratifying oncologic patients.

## Figures and Tables

**Figure 1 cancers-13-00591-f001:**
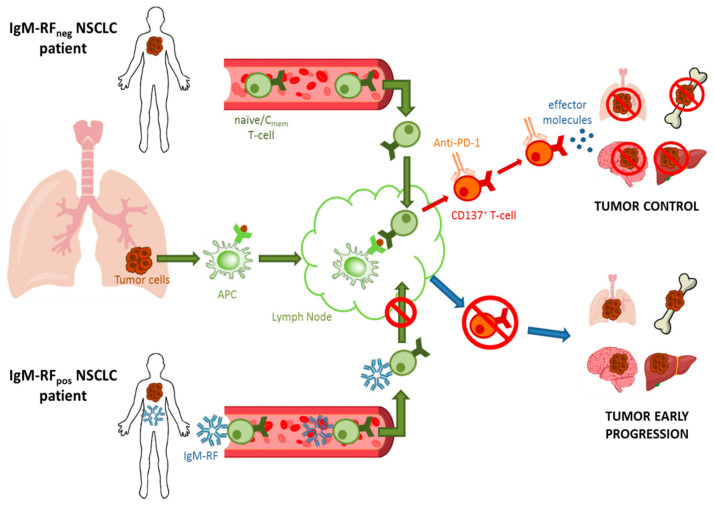
Schematic representation of the suggested effect of immunoglobulin class M (IgM)-rheumatoid factor (RF) in limiting naïve and central memory T-cells recirculation in non-small lung cancer (NSCLC) patients, leading to an impaired CD137^+^ tumor-directed T-cells expansion and a consequent failure of immune-based immunotherapies.

**Figure 2 cancers-13-00591-f002:**
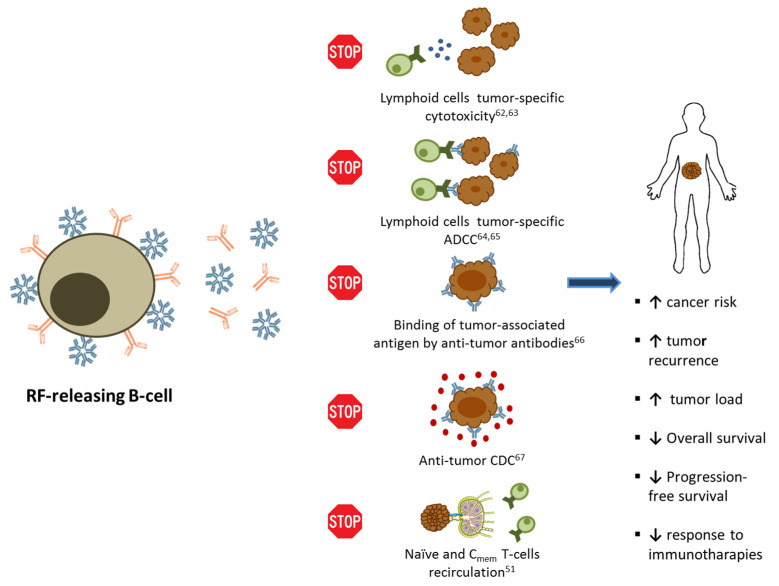
Schematic representation of the different in vitro immune-suppressive activities of RFs in the context of anti-tumor responses with their effects on cancer patients.

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
