# Peer review of "Rheumatoid Factor: A Novel Determiner in Cancer History"

_cancers, 2021, doi:10.3390/cancers13040591_

Round 1
Reviewer 1 Report
The topic is quite interesting and not widely discussed previously, which is the strength of the manuscript. the organization is overall adequate but has some correctable flaws. The English phrasing is generally understandable but frequently awkward with unusual and uncomfortable phrasing. The manuscript would benefit from careful and extensive editing by an immunologist who is a better writer in English.
Organizationally it would benefit the manuscript to point out the role of RF in both limiting the antitumor effects of antibodies, whether endogenous or therapeutic, and what appears to be a distinct potential impact on effector T cells. the impact on antibodies is obvious, the impact on T cells is not and that should be highlighted and discussed in a bit more depth.
the abstract could be expanded to include more information and the word "clinchers" is poorly used and unclear.
The figures are nicely drawn and well organized but both would benefit from explanatory legends and the fonts in figure 2 could be increased to make it easier to read. there is plenty of empty space to accommodate that change.
lines with awkward or unclear phrasing include but are not limited to these
24,30,36,39,54,63,91,96,120,130,135,148
lines 97,98 are not clear
line 115: early progression phrase is not clear, early progression of what?
line 70,71 has a statement that was similarly stated in the previous paragraph and adds little to be restated
Author Response
- The topic is quite interesting and not widely discussed previously, which is the strength of the manuscript. the organization is overall adequate but has some correctable flaws. The English phrasing is generally understandable but frequently awkward with unusual and uncomfortable phrasing. The manuscript would benefit from careful and extensive editing by an immunologist who is a better writer in English.
Response. English has been extensively revised.
- Organizationally it would benefit the manuscript to point out the role of RF in both limiting the antitumor effects of antibodies, whether endogenous or therapeutic, and what appears to be a distinct potential impact on effector T cells. the impact on antibodies is obvious, the impact on T cells is not and that should be highlighted and discussed in a bit more depth.
Response. We thank the reviewer for this useful suggestion. We have better clarified this point in section 4 of our manuscript and added the following paragraph to the discussion:
“Altogether, these findings suggest that high titers of RF in patients’ sera could inhibit the anti-tumor immunity in different ways (Fig. 2).
In fact, RFs’ physiological role of limiting a redundant immune response against pathogens seems coherent with their potential immunosuppressive activity within anti-tumor immune responses. On one side, in healthy conditions, it has been pointed out how RFs are able of facilitating the clearance of immune-complexes and antibodies. Similarly, in cancer, their ability of binding antibodies results in interfering with the anti-tumor effect of both endogenous and therapeutic antibodies. It has indeed been demonstrated that RFs can hamper the interaction between tumor-directed antibodies and antigens on cancer cells surface as well as impair antibodies-mediated anti-tumor effector functions, such as CDC or ADCC. On the other side, the physiological effect exerted by RFs on effector T-cell has not been clarified yet. However, from different works presented in this review, it emerged that in the context of anti-cancer immune responses the presence of RFs is able to impair lymphocytes-mediated anti-tumor cytotoxicity and the recirculation of naïve and central memory T-cells, leading to a reduced expansion and effectiveness of tumor-specific effector T-cells. Therefore, in the light of these evidence, it is reliable to assume that RFs presence in the blood of cancer patients could facilitate cancer growth and progression”.
- the abstract could be expanded to include more information and the word "clinchers" is poorly used and unclear.
Response: the abstract has been completely rewritten as follows:
“Abstract: The possible interplay between autoimmunity and cancer is a topic that still needs to be deeply explored. Rheumatoid Factors are autoantibodies able of binding the Fc region of IgGs. In physiological conditions, their production is a transient event aimed at contributing in the elimination of pathogens as well as limiting a redundant immune response, by facilitating the clearance of antibodies and immune-complexes. Their production can become persistent in case of different chronic infections or diseases, being for instance a marker for the diagnosis and prognosis of Rheumatoid Arthritis. Their presence is also associated with aging. Some studies then highlighted how elevated levels of RFs in the blood of patients are correlated with an increased cancer risk, tumor recurrence and load and with a reduced response to anti-tumor immunotherapies. In line with their physiological roles, RFs showed in different works the ability of impairing in vitro anti-cancer immune responses and effector functions, suggesting their potential immunosuppressive activity in the context of tumor immunity. Thus, the aim of this review is investigating the emerging role of RFs as determiners of cancer faith”.
- The figures are nicely drawn and well organized but both would benefit from explanatory legends and the fonts in figure 2 could be increased to make it easier to read. there is plenty of empty space to accommodate that change.
Response: Figure legends have been added and figure 2 has been modified according to reviewer’s suggestion.
- lines with awkward or unclear phrasing include but are not limited to these: 24,30,36,39,54,63,91,96,120,130,135,148
Response. We have revised the English throughout the manuscript and corrected the cited sentences.
- lines 97,98 are not clear
Response. The entire period has been rewritten as follows:
“This association was investigated for the first time in the Reykjavik area (Iceland) where, starting from 1967, a general health survey was conducted[53,54]. Then, the women that were tested positive for RF were divided in groups based on RF titres and followed up until 1974: of the 4 women that died during this observation period, 3 belonged to the group with the highest RF titres; all of these 3 women had been diagnosed with cancer (2 mammalian cancer and 1 lung cancer). Thus, it was suggested that high blood concentrations of RFs in healthy subjects might be associated with an increased risk of developing cancer[53,54]”.
- line 115: early progression phrase is not clear, early progression of what?
Response. The sentence has been corrected as follows:
“Finally, a retrospective study conducted in our laboratory brought a clear evidence of how the IgM-RF positivity is a strong predictive factor for the development of NSCLC patients early progression in response to the treatment with anti-PD-1 Immune-Checkpoint inhibitors (ICIs)”.
- line 70,71 has a statement that was similarly stated in the previous paragraph and adds little to be restated
Response. The sentence has been summarized and merged with the previous one in order not to be redundant.
Reviewer 2 Report
The manuscript provides readers with an overview of the clinical problem and contains update information regarding the roles of Rheumatoid Factor in tumor history and cancer Immunotherapy response. The review is well written and is suitable for publication following minor revisions of the English language. Summarizing figures are really helpful to understand the key points of the manuscript.
However there is an issue on which I would like to make a comment:
- the authors should deepen, in the conclusion section, the clinical utility of RF as prognostic and/or predictive factor.
Author Response
The manuscript provides readers with an overview of the clinical problem and contains update information regarding the roles of Rheumatoid Factor in tumor history and cancer Immunotherapy response. The review is well written and is suitable for publication following minor revisions of the English language. Summarizing figures are really helpful to understand the key points of the manuscript.
Response. English has been revised.
However there is an issue on which I would like to make a comment:
- the authors should deepen, in the conclusion section, the clinical utility of RF as prognostic and/or predictive factor.
Response: We thank the reviewer for pointing this out. We have added this paragraph in the conclusion section:
“These findings about the possibility of using RF as both prognostic and predictive factor in cancer patients, look promising from two different perspectives. First, being now clear that RF can be used as an indicator of an heightened susceptibility of developing cancer as well as of an increased tumor recurrence, individuals that are positive for RF could be monitored more strictly so to increase early diagnosis. Second, since the presence of RF in cancer patients correlates with a reduced OS and PFS and a lack of response to anti-cancer immunotherapies, the RF dosage could be introduced in oncologic patients together with other parameters, in order to improve their risk stratification and to help adjusting the therapeutic strategy based on the single patient’s characteristics. Being RF dosage a routinely performed and already standardized test, this seems to be a promising opportunity to aid clinicians that are struggling to predict patients faith and, above all, to increase the number of responder patients to anti-cancer therapies”.